# Decoding Bacterial Motility: From Swimming States to Patterns and Chemotactic Strategies

**DOI:** 10.3390/biom15020170

**Published:** 2025-01-23

**Authors:** Xiang-Yu Zhuang, Chien-Jung Lo

**Affiliations:** 1Department of Physics and Center for Complex Systems, National Central University, Zhongli, Taoyuan 32001, Taiwan; 2Institute of Physics, Academia Sinica, Taipei 115201, Taiwan

**Keywords:** bacterial flagellar motility, flagellar polymorphism, swimming state, swimming pattern

## Abstract

The bacterial flagellum serves as a crucial propulsion apparatus for motility and chemotaxis. Bacteria employ complex swimming patterns to perform essential biological tasks. These patterns involve transitions between distinct swimming states, driven by flagellar motor rotation, filament polymorphism, and variations in flagellar arrangement and configuration. Over the past two decades, advancements in fluorescence staining technology applied to bacterial flagella have led to the discovery of diverse bacterial movement states and intricate swimming patterns. This review provides a comprehensive overview of nano-filament observation methodologies, swimming states, swimming patterns, and the physical mechanisms underlying chemotaxis. These novel insights and ongoing research have the potential to inspire the design of innovative active devices tailored for operation in low-Reynolds-number environments.

## 1. Introduction

A cell as small as a bacterium is constantly buffeted by Brownian motion in aqueous environments. To achieve active movement, many bacterial species are equipped with a specialized propulsion apparatus, the bacterial flagellum. This system consists of a long extracellular filament approximately 20 nm in diameter, a hook about 50 nm in length, and rotary bacterial flagellar motors (BFM) embedded in the cell envelope. The flagellum enables bacteria to move actively, surpassing the diffusion limits of attractants or repellents, facilitating single-cell motility and chemotaxis [1,2,3]. Bacterial locomotion and signal transduction systems are interconnected and essential for a variety of activities, including nutrient foraging, infection, colony expansion, biofilm formation, and establishing symbiotic relationships. The intricate chemotaxis system coordinates sophisticated responses through interactions among chemoreceptors, which act as sensory tools: the signal transduction pathway and the dynamic BFM.

The bacterial flagellum is more than a simple propulsion device; it exhibits dynamic and complex behavior influenced by factors such as motor rotational states, flagellar polymorphisms, and variations in flagellar arrangement and configuration [4] (Figure 1). This complexity allows bacteria to swim in different modes, executing sequences of straight runs interspersed with turning events, resulting in swimming patterns that mimic the effective diffusion of active particles [4]. Chemotactic behavior is a highly dynamic process in which cells modulate flagellar motor rotation in response to environmental cues detected by chemoreceptors. The underlying chemotactic behavior is a dynamic process wherein cells modulate flagellar motor rotation states in response to environmental information gathered by their chemoreceptors. This modulation is achieved by orchestrating different motility states through flagellar motor rotation (Figure 1). A classical example is *Escherichia coli*, which uses motor switching to alternate between run and tumble states [5]. In contrast, *Vibrio alginolyticus* employs a push–pull–flick pattern, enhancing its search efficiency [6,7,8]. Recent discoveries, such as the flagellar wrap mode, have expanded our understanding of bacterial motility, revealing its role in obstacle evasion [9], surface gliding [10], and chemotaxis enhancement [11,12].

To unravel the complexities of bacterial motility in dynamic environments, it is essential to visualize flagellar filament behavior in real time. Recent advances in flagellar filament labeling and imaging techniques have provided new insights into the intricate mechanisms of bacterial motility and chemotaxis. In this review, we first summarize the experimental techniques and diverse swimming patterns that have broadened our understanding of chemotaxis. We then decompose swimming states into distinct state conditions and provide an overview of various swimming patterns (Figure 1). Finally, we discuss how bacteria utilize these patterns to navigate complex environments through simple random walks and biased random walk modulation. These sophisticated behaviors emerge from the fundamental physical mechanism of two-state modulation.

## 2. Real-Time Imaging of Fluorescent Flagellar Filaments

### 2.1. Fluorescence Labeling

The flagellar filament is a hollow tube 5–20
μm in length, with typical outer and inner diameters of 24 nm and 2 nm, respectively [13]. Due to its slender dimensions, standard optical bright-field microscopy cannot clearly resolve individual flagella. The first pioneering real-time fluorescence imaging of individual flagella was applied to swimming *E. coli* without impairing bacterial motility [14]. This technique uses succinimide (NHS) esters conjugated with specific Alexa Fluor dyes, which covalently react with primary amine (R-NH2) groups at the N-terminus and lysine residues of proteins on the outer surfaces of the flagellar filament and cell body (Figure 2A, box a).

A similar yet more effective approach was later developed using Thiol-reactive maleimide dyes. These dyes leverage a combination of genetics and click chemistry to form covalent bonds with free sulfhydryl (-SH) groups on cell surface proteins. This method is milder and more specific than NHS ester tagging, as sulfhydryl groups are less abundant than primary amines. The number of sulfhydryl groups can be controlled by introducing or removing non-disruptive cysteine residues in target proteins. The positioning of these cysteine residues is crucial, as it significantly affects resolution and signal strength. To preserve protein folding and maintain nucleophilic properties, cysteine is often substituted with serine or threonine. Maleimide dyes have been employed to label flagella, providing insights into bacterial motility [9,12,15,16] and flagellar growth dynamics [17,18,19]. Although this technique requires specific conditions and extended staining periods, it enables temporal studies of cellular structures through sequential staining protocols. Furthermore, unlike NHS ester dyes, maleimide dyes are compatible with Gram-positive strains without inducing toxic side effects.

An alternative, highly specific approach involves the use of optimized tetracysteine tags (TC-tags) inserted into flagellin, enabling the dynamic labeling and tracking of single-flagellum growth [20]. This enhanced TC-tag has increased binding affinity for diarsenic dyes, such as FlAsH and ReAsH, allowing for the high-resolution fluorescence imaging of extracellular nanostructures without the need to wash out low-fluorescent, unbound reagents (Figure 2A, box b). This facilitates the real-time visualization of flagellar assembly dynamics, providing valuable insights into the mechanisms underlying flagellar formation.

A subgroup of bacteria possess sheathed flagella, in which the flagellar filaments are covered by a membrane-like lipid layer. This sheath prevents fluorescent molecules from binding directly to flagellar proteins. However, the use of lipophilic dyes for filament labeling can transform this limitation into an advantage. This characteristic allows lipophilic fluorescent dyes, such as NanoOrange or FM lipophilic styryl dyes, to target hydrophobic membranes (Figure 2A, box c), enabling high-resolution fluorescence imaging without the need for intermittent washing steps [21,22,23]. Since lipophilic dyes do not form covalent bonds, their interaction with lipids occurs rapidly, allowing for the direct visualization of growing filaments [24]. This method has been applied to the single polar flagellum of *V. alginolyticus*, enabling the real-time measurement of flagellar growth [21].

### 2.2. Real-Time Observation

Fluorescence labeling significantly enhances the contrast in flagellar filament imaging, enabling detailed single-filament observation. Wide-field epifluorescence microscopy is a widely employed and straightforward technique for bacterial studies. It provides sufficient spatial resolution to capture bacterial flagella. During image acquisition, a broad cone of excitation light at a specific wavelength uniformly activates all fluorophores within a large 3D volume (Figure 2B, left). While this method reflects the natural conditions of free-swimming bacteria, the rapid rotation of flagella at several hundred hertz complicates the acquisition of clear images of flagellar configurations.

Total internal reflection fluorescence (TIRF) microscopy is a powerful technique for studying dynamic molecular events near surfaces. Unlike conventional fluorescence microscopy, TIRF selectively excites fluorophores located close to the surface by generating an exponentially decaying evanescent wave through illumination at or above the critical angle (Figure 2B, right). This approach ensures a high signal-to-noise ratio and superior z-axis resolution without compromising temporal resolution. Several studies have demonstrated that TIRF microscopy is particularly effective for investigating highly dynamic surface-associated bacterial motility [10], as well as processes involving cytoskeletal factors and peptidoglycan synthases.

Capturing the images of free-swimming bacteria in their natural 3D environments requires a large field of view (several hundred micrometers) and high-speed imaging to track rapidly rotating flagella at kilohertz frequencies. While sCMOS cameras can achieve frame rates in the kilohertz range, this often reduces the region of interest (ROI) and increases the risk of phototoxicity. Standard video-rate acquisition results in blurred flagellar images due to overexposure during multiple rotations (Figure 2C). A useful alternative is strobe illumination, which enables the visualization of flagellar motion without compromising the field of view [14,25]. In this method, the excitation laser light is strobed using a slotted wheel and synchronized with the camera. Strobe illumination can also be achieved using an LED light source with flexible control and camera synchronization [26]. The core principle is to shorten the fluorescence excitation to milliseconds, effectively “freezing” transient flagellar motion while maintaining a large field of view (Figure 2C). This approach allows for the observation of filament dynamics in free-swimming bacteria.

## 3. Swimming States

Micron-sized bacteria cannot directly steer toward target areas due to the limited spatial resolution of their chemical sensing. As a result, their swimming trajectories consist of several straight runs interspersed with angular changes, producing a random-walk-like motion. While the bacterial flagellar motor is structurally similar across species, various factors influence the duration of runs and the angles of directional changes, which differ between species. In this section, we will dissect the factors that govern bacterial swimming states.

### 3.1. Flagellar Motor Rotational States

The real engine is the bacterial flagellar motor powered by the electrochemical energy of selected ions. The BFM can rotate either clockwise (CW) or counterclockwise (CCW) [27,28,29,30] (Figure 3A), and the switching is linked to the chemotaxis behavior by the signal transduction pathway. For instance, *E. coli* and *Salmonella* detect chemotactic gradients of attractants and repellents through methyl-accepting chemotaxis proteins (MCPs), which are transmembrane chemoreceptors located at cellular poles [31]. When ligands bind to MCPs, CheA is activated and undergoes autophosphorylation. The phosphorylated CheA (CheA-P) then transfers its phosphate group to the response regulator CheY, producing phosphorylated CheY (CheY-P). CheY-P binds to the flagellar motor, modulating the rotation direction of the flagella and influencing bacterial movement [32,33,34]. Meanwhile, CheZ catalyzes the dephosphorylation of CheY-P, terminating the signal and resetting the chemotaxis system to prepare for the next round of signal transduction [35,36]. Phosphorylated CheY then binds to FliM and FliN, switching the rotation direction from CCW to CW [1,37,38]. Cryo-electron microscopy has revealed that the C-ring diameters are 416 Å for CCW and 407 Å for CW, indicating that the C-ring subunits are closer together in the CW state [39,40,41]. Additionally, the conformation of FliG-FliM is altered by the binding of CheY-P to the C-ring, inducing an angular change that causes the C-ring to switch between CCW and CW [42]. Consequently, when receptors sense temporal changes in extracellular stimuli, the intracellular chemotactic signaling network sends a cascade of signals to the flagellar motor. By switching between CCW and CW, the flagellar motor adjusts its pathway strategy. Moreover, sensory receptors can detect other factors such as temperature and pH.

In addition to the CCW and CW rotational states, the BFM has other possible STOP states. For instance, the molecular clutch in the *Bacillus subtilis* BFM is a mechanism that allows bacteria to switch off motility during the biofilm formation process [43]. This clutch involves a protein called EpsE, which interacts with the flagellum’s rotor protein FliG, causing a conformational change that disengages the rotor from the proton-powered stator (Figure 3A). This process is rapid, simple, and reversible, allowing bacteria to efficiently halt or resume movement without the need for new flagellar synthesis. Another STOP state is a molecular brake mechanism reported in *Rhodobacter sphaeroides* [44]. Unlike most bacterial species that use bidirectional BFMs, *R. sphaeroides* employs a single stop–rotate BFM. The stop state is achieved by locking the rotor in place with high torque. In conclusion, there are three possible states of the BFM rotation: CCW/CW/STOP.

### 3.2. Flagellar Polymorphism

The bacterial flagellar filament is a helical propeller composed of thousands of copies of the protein flagellin (FliC). The filament is a supercoiled assembly formed by 11 protofilaments. During bacterial swimming, the reversal of motor rotation switches the filament between left-handed (LH) and right-handed (RH) supercoils. These dynamic polymorphic transformations arise from combinations of two distinct conformations and packing interactions of the L-type and R-type protofilaments. X-ray diffraction reveals a 0.8 Å intersubunit distance difference between R-type and L-type protofilaments [45,46]. Therefore, there are two straight forms of flagellar filaments (all L-type or all R-type) and ten helical forms composed of combinations of L-type and R-type protofilaments (Figure 3B). Several helical polymorphic shapes have been identified [47,48,49,50,51]. The dynamic polymorphic transformation during swimming, driven by motor switching, produces many swimming patterns [14,26]. Flagellar polymorphism is influenced by factors such as the 3D structure of flagellin, ionic strength, and loading force [29].

### 3.3. Flagellar Arrangement

The arrangement of flagella on the cell surface plays a crucial role in motility and chemotaxis and can be classified into five major groups (Figure 3C) [52]. Monotrichous bacteria, such as *V. alginolyticus*, have a single polar flagellum at one end. Lophotrichous bacteria, including *Aliivibrio fischeri* and *Pseudomonas putida*, possess multiple flagella at one pole. Amphitrichous bacteria, such as *Campylobacter jejuni*, have a single flagellum at both ends. Bipolar lophotrichous bacteria exhibit multiple flagella at both poles. Peritrichous bacteria, such as *E. coli* and *S. enterica*, display multiple flagella distributed around the cell body [14,53]. The flagellar arrangement affects the bacterial flagellar configuration and consequently influences swimming patterns. Further details will be elaborated in the following sections.

### 3.4. Flagellar Configuration

In addition to flagellar arrangement, the flagellar filament can adopt various configurations due to the flexibility of the hook and flagellar filament polymorphism. These configurations include tailed, split, spread, flick, and wrap states (Figure 3D). The tailed and wrap configurations typically result in a run swimming mode, while split, spread, and flick configurations correspond to directional changes during swimming. A tailed configuration features a filament or a bundle of filaments at one end of the cell body (Figure 3D), resembling a propeller at the stern. In bacteria with multiple flagella, motor switching may lead to a split flagellar configuration (Figure 3D). In extreme cases, all flagellar filaments separate from the bundle, resulting in the spread configuration (Figure 3D). Some bacteria, such as *V. alginolyticus*, exploit flagellar buckling instability to rapidly change direction [6,7]. Hook buckling triggers flick events, inducing abrupt direction changes. Additionally, flagellar filaments display complex dynamics influenced by motor switching, generating significant torque that twists the filaments and alters chirality between the RH and LH states. Recently, a novel wrapping mechanism was identified [52], adding to the complex nature of bacterial motility.

### 3.5. Swimming States

The primary swimming modes are push, pull, tumble, flick, and wrap. These actions are the combined result of the direction of flagellar rotation, filament configuration, filament polymorphism, and arrangement of flagella on the cell surface (Figure 3E). For example, peritrichous *E. coli* exhibits a ’run’ (push) state when it is in the CCW/LH/tailed configuration. The cell enters the ‘tumble’ mode by switching the rotation of one or more motors to CW, changing some filaments to RH, and producing a split configuration (CW/RH/split). Monotrichous *V. alginolyticus* exhibits push and pull modes in the (CCW/LH/tailed) and (CW/LH/tailed) states, respectively. It can enter the ’flick’ mode in the (CCW/flick/LH) state. The wrap state in polar flagellated bacteria is characterized by the filament wrapping around the cell body when the cell is in the (CW/LH/wrap) configuration. Despite its unique structure, the swimming direction remains consistent with the polar flagella pull mode, categorizing the push–wrap pattern as a push–pull movement.

In the run modes, thrusting force directions result from several state parameters, including motor rotation direction, flagellar configuration, and flagellar handedness (Figure 3F). The motor rotation direction switch is the trigger, but it can lead to various states depending on the mechanical properties of the hook and filament.

## 4. Swimming Patterns

Bacteria perform chemotaxis through swimming patterns that consist of different swimming states. These swimming states are influenced by parameters such as flagellar rotation, filament polymorphism, flagellar arrangement, and configuration. Together, these factors result in a diverse array of swimming patterns that enhance the adaptive capabilities of bacteria in their environments. In this section, we will provide a comprehensive discussion of the three main swimming patterns.

### 4.1. Run-and-Tumble

*E. coli*, characterized by a peritrichous flagellar arrangement, serves as a classic model for studying flagellar motility and chemotaxis. Employing the well-known ’run-and-tumble’ pattern in response to chemotaxis (Figure 4A), *E. coli* typically has approximately 4–8 left-handed flagellar filaments per cell, which form a bundle when the flagellar motor rotates counterclockwise (CCW/LH/tailed). This state, termed the ’run’ mode, enables the bacteria to advance. Conversely, when the flagellar motor switches from counterclockwise to clockwise rotation, the flagellar filaments undergo a polymorphic transition from left-handed to right-handed (CW/split/RH) (Figure 4A). As a result, the right-handed filaments detach from the bundle, inducing a ’tumble’ motion that facilitates bacterial reorientation. The run-and-tumble phenomenon underscores that flagellar motion is accompanied by polymorphic transitions in flagellar filaments due to their flexible and transformable nature [5,54,55]. By modulating the duration of the ’run’ and ’tumble’ phases, *E. coli* actively navigates toward favorable sources or away from repellents in biased random walks [5,14,52,56,57,58].

The polymorphism of *E. coli*’s flagella can be described in more detail. The filaments are bundled in the normal left-handed (LH) polymorphism while the cell is in the run mode. When the flagellar motor switches from counterclockwise to clockwise rotation, the affected flagellum leaves the bundle, transitioning from a left-handed ’normal’ state to a right-handed ’semi-coiled’ or ’curly’ state. *E. coli*’s swimming pattern combines flagellar polymorphic changes, following a process that ensures brief and efficient tumbling that enables bacteria to explore their environment optimally to perform chemotaxis [59].

As an additional example, *R. sphaeroides*, a monotrichous bacterium that behaves similarly to *E. coli*, exhibits a normal flagellar shape when swimming forward. However, upon motor rotation stop and reorientation, its flagellum assumes a ’coiled’ configuration during a stationary state [60,61].

### 4.2. Push–Pull–Flick

Various bacterial species have developed distinct mechanisms for motility and chemotactic responses, driven by differences in flagellar arrangements and filament properties that deviate from the tumbling observed in *E. coli* [52,57,62]. Another prevalent swimming pattern among polar flagellar bacteria is the ’run–reverse–flick’ or ’push–pull–flick’ pattern (Figure 4B), initially identified in *V. alginolyticus* [6,7].

In an aqueous environment, *V. alginolyticus* is propelled by a left-handed filament covered with a membrane-like sheath structure. The counterclockwise rotation of *V. alginolyticus*’s flagellar filament generates forward thrust (CCW/LH/tailed), while a switch to clockwise rotation results in a pulling motion (CW/LH/tailed), causing the bacterium to swim in reverse. When the motor switches back to CCW rotation, the mechanical failure of the hook, known as buckling, leads to a 90-degree reorientation of the flagellar filament, similar to thrust vectoring. This ’flick’ mode (CCW/LH/flick) is attributed to the instability of the flagellar hook, which is dependent on the bacterial size and swimming speed [7,63].

Remarkably, the chemotactic behavior of *V. alginolyticus* differs significantly from that of *E. coli*. When stimulated, *V. alginolyticus* amplifies the disparity between forward and backward durations and prolongs CCW rotation when swimming in a favorable direction [6]. *V. alginolyticus* can thus bias its swimming direction very quickly [64,65]. Consequently, the ’push–pull–flick’ pattern of *V. alginolyticus* proves more efficient in chemotaxis than the ’run-and-tumble’ pattern observed in *E. coli* [6].

### 4.3. Wrap It Up

The first evidence of the wrap mode can be traced back more than 100 years to dark-field microscopy observations of lophotrichous bacterial motility. In these early observations, the filament bundle folded down to the cell body during the push–pull transition [52]. However, the dominant light scattering from the cell body prevented the clear observation of the filament configuration. The direct observation of wrap events was achieved only with the recent development of flagellar filament fluorescence labeling and microscopy (see Section 2). It is noteworthy that flagella at the leading pole of amphitrichous *Magnetospirillum magneticum* do not point away from the cell (pull mode) but rotate around the cell (wrap mode) [66]. Several research groups later identified the wrap mode in monotrichous *Shewanella putrefaciens* [9] and *P. aeruginosa* [12] and lophotrichous *P. putida* [16], *Burkholderia* sp. [10], and *A. fischeri* [26] cells. It is now believed that the wrap mode is not exceptional but rather a general swimming mode for flagellated bacteria.

#### 4.3.1. Swimming Patterns with Wrap Mode

Several polar flagellated bacteria undergo a transition in flagellar configuration, shifting from a ’push’ to a ’wrap’ mode through an intermediate ’pull’ state. During motor rotational switching, the flagellar filaments wrap around the bacterial body, allowing the bacterium to continue swimming. When altering swimming direction, the wrapped filaments can extend away from the cell, demonstrating a reversible process. This wrapping becomes prominent in *S. putrefaciens* when encountering resistance or highly viscous environments [9]. Under high-load conditions, clockwise rotation destabilizes the flagellar filament, triggering a polymorphic transition into straight filaments. These filaments are then rewound around the bacterium (Figure 5A). In *P. aeruginosa*, flagellar wrapping is observed during the transition from ’pull’ to ’push’ modes, but not during the reverse process [12]. The reversible helical wrapping is linked to the buckling of the hook region (Figure 5B). Overall, the wrap mode introduces an additional swimming pattern, expanding the motility repertoire.

#### 4.3.2. Wrapping Mechanism

Recent observations of the flagellar wrap mode have raised intriguing questions about its causes and implications. Based on direct observations, it has been proposed that wrapping may be caused by flagellar instability during motor rotation reversal and shifts in applied torque. This wrapping could function as an escape mechanism in complex environments [9]. A subsequent mathematical model, developed from observations of *P. putida*, explored the dynamics of lophotrichous bacteria [67] (Figure 5A). The model identified critical torque thresholds that delineate transitions between wrap and pull modes during clockwise (CW) motor rotation, and between push and flick-like modes during counterclockwise (CCW) rotation. The study highlighted the importance of transitions between wrap and push modes—along with pauses—in determining new swimming paths. The degree of the reorientation of the swimming direction depended on the length of motor stoppage between periods of motor rotation. Additionally, wrap mode was suggested to facilitate escape from confined regions, particularly near surfaces. Interestingly, the model indicated that if the bending moduli of both the filament and hook are sufficiently high, the bacterium predominantly exhibits either pull or push modes, depending on the direction of motor rotation. However, when the flagellum is overly flexible, the motor’s torque is not effectively transmitted to the filament, potentially disrupting sustained swimming modes.

Park et al.’s numerical simulation assumed a single polar flagellum with an intrinsically left-handed helix. However, recent findings by Tian et al. revealed that the filament of *P. aeruginosa* assumes a right-handed helix in the wrap mode, unlike its typical left-handed chirality in pull or push modes [12] (Figure 5B). This suggests that the polymorphic transition of flagella in *P. aeruginosa* is induced by hook buckling instability during motor switching, akin to the flick mode observed in *V. alginolyticus* [7]. The bending stiffness of *P. aeruginosa*’s hook in its relaxed state was measured at EI =
$1.27\times 10^{-26}$ Nm^2^, approximately one-third of the value reported for *V. alginolyticus* [7,12]. This disparity in chirality indicates that when flagella are sufficiently soft and synchronized, fluid flow exerts torque at the filament base, prompting a shift in chirality [68]. As a result, variations in chirality, rotational direction, and flagellar positioning contribute to the diverse swimming behaviors observed in bacteria.

#### 4.3.3. Wrap for Extending Run

Recently, a novel type of dynamic polymorphic transition of the filament during wrap mode has been identified [26]. In highly viscous mucus environments, bacteria experience reduced swimming speeds, which consequently diminish overall motility and limit the range of chemotactic searches. As *A. fischeri* navigates in search of a symbiotic partner, it encounters significant challenges due to high viscosity and a restricted target area. Interestingly, *A. fischeri* has evolved a viscosity-dependent swimming pattern as an adaptive strategy. In low-viscosity environments, *A. fischeri* primarily employs a push–pull swimming pattern. However, as viscosity increases, motor switching generates substantial twisting and dragging forces, triggering a flagellar polymorphic transition from push to wrap mode (Figure 6A). During the wrap mode, filament transformations induced by motor switching extend backward swimming to generate long runs (Figure 6B). If the synchronization of motor rotation is disrupted, the intertwined flagella spread apart, resembling the tumbling behavior of Escherichia coli (split/spread configuration). Notably, the transition from push to wrap mode may also occur through this spread configuration (Figure 6A). Overall, the motility pattern of *A. fischeri* incorporates all known bacterial swimming behaviors, except for flicking (Figure 6A).

Many bacteria encounter highly viscous environments such as mucus or surfaces. *V. alginolyticus* has evolved a secondary, proton-driven lateral flagellar system for surface swarming [69]. When sensing increased viscous drag, *V. alginolyticus* produces lateral flagella to aid in motility [70]. For bacteria that lack a secondary flagellar system, wrapping behavior serves as an alternative motility strategy in high-drag environments. This novel wrap mode allows bacteria to maintain a larger search area in high-viscosity environments. By wrapping flagellar filaments around the cell body, bacteria can move backward and escape from constricted spaces [9]. During this backward movement, the filament’s waveform remains unchanged relative to the surface, enabling smooth backward progression along the helix. In contrast, when viscosity increases in the absence of a solid surface, the flagellar helix’s waveform shifts relative to the medium. This shift can hinder the efficient translation of the cell body. These observations suggest that flagellar wrapping and screw-like movement can help cells escape narrow channels, promoting more efficient navigation through confined environments [67,71].

## 5. Discovered Bacterial Swimming States and Transitions

In response to environmental changes, such as nutrient gradients, bacteria dynamically adjust flagellar motor rotation between counterclockwise (CCW) and clockwise (CW) to perform random and biased random walks [5] (Figure 1). The probability of CW or CCW rotation is influenced by external chemical concentrations and the cell’s internal signaling network [58]. The chemotaxis system, which integrates chemoreceptors, signal transduction pathways, and locomotion apparatus, directs bacterial movement (Figure 1).

Bacterial swimming patterns consist of distinct states. For example, during forward swimming (run), *E. coli* flagella rotate CCW, forming a cohesive bundle (Figure 4A). Upon sensing a decreasing chemoattractant gradient, phosphorylated CheY (CheY-P) accumulates and binds to the motor’s C-ring, increasing the probability of CW rotation. This induces a tumble, causing the cell to reorient randomly (Figure 4A). Conversely, in the presence of an increasing attractant gradient, methyl-accepting chemotaxis proteins (MCPs) inhibit the production of CheY-P, sustaining CCW rotation and allowing *E. coli* to continue running toward higher attractant concentrations. Through this interplay, *E. coli* employs both random and biased random walks to achieve chemotaxis (Figure 1). Hence, bacterial chemotaxis is closely tied to flagellar motor dynamics and swimming patterns.

Over extended periods, the mean square displacement (MSD) of the bacterial cells remains linear with time, allowing the calculation of an active diffusion coefficient [4,58,72]. This coefficient
$$D=\frac{V^{2}\tau }{3(1-\alpha )},$$ evaluates bacterial motility, where *V* is the swimming speed,
τ is the mean duration of straight runs, and
α represents the mean cosine of the angle between successive runs. For *E. coli*’s run-and-tumble pattern,
α is 0.33 [4,5]. For run-and-reverse (push–pull or push–wrap) patterns,
α = −1. For the push–pull–flick pattern,
α is −1 and 0 [73]. Variations in swimming patterns influence efficiency and adaptability of bacterial navigation. For bacteria exhibiting complex swimming patterns with multiple run modes, further analytical studies and direct measurements of the active diffusion coefficient are essential to fully characterize their motility.

In Newtonian aqueous media, bacteria typically follow ideal swimming patterns. However, diverse fluid environments—ranging from crystalline mucus to structured soils [74,75,76,77,78]—introduce significant complexity. Observations in complex fluids, such as polymer solutions or colloidal suspensions, reveal enhanced bacterial mobility through trajectory straightening and reduced tumbling [77], as well as the emergence of new swimming patterns resembling slow random walks [78]. These phenomena are driven by intricate interactions between motile bacteria and complex fluids, influenced by effective fluid properties or inhomogeneities at scales comparable to the scale of bacterial cells or flagella [79].

The known swimming states and their possible transition routes are summarized in Figure 7 and Table 1. Notably, bacteria exhibit multiple run- and angle-changing states. Certain transitions are irreversible, driven by the mechanical properties of the hook or flagellar filaments. Considering the complexity and adaptive value of bacterial motility, it is likely that additional swimming states await discovery.

## 6. Conclusions

As more patterns of bacterial motility are discovered and characterized (Table 1), a detailed state transition map can be constructed (Figure 7). This comprehensive map allows bacteria to optimize motility states for exploration within specific structural environments, enhancing adaptability and survival across diverse conditions. State transitions result from the mechanical and chemical properties of the hook and flagellar filaments. Further analysis is necessary to fully understand these transition routes and their constraints. Insights from this map, along with knowledge of flagellar polymorphic transitions, hook instability, and motor synchronization, could inform the design of novel microscopic active agents, such as genetically engineered bacteria optimized for environmental or medical applications, or miniature robotic devices mimicking bacterial motility to navigate complex microenvironments for tasks like drug delivery or environmental remediation.

## Figures and Tables

**Figure 1 biomolecules-15-00170-f001:**
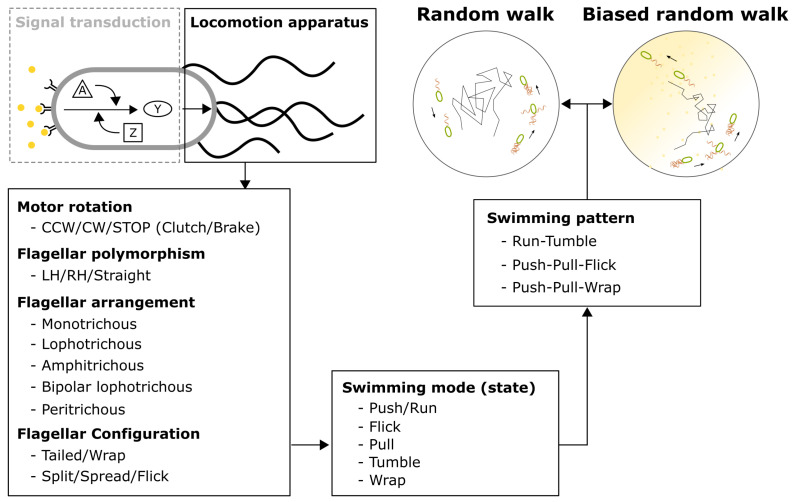
The complexity of bacterial motility and the simplicity of chemotaxis behavior. Bacterial chemotaxis integrates signal transduction with locomotion. Locomotion is influenced by motor rotation, flagellar polymorphism, flagellar arrangement, and configuration. These factors collectively determine specific swimming modes, while a swimming pattern consists of a series of such modes. Bacteria modulate the states within a swimming pattern to perform random walks and biased random walks.

**Figure 2 biomolecules-15-00170-f002:**
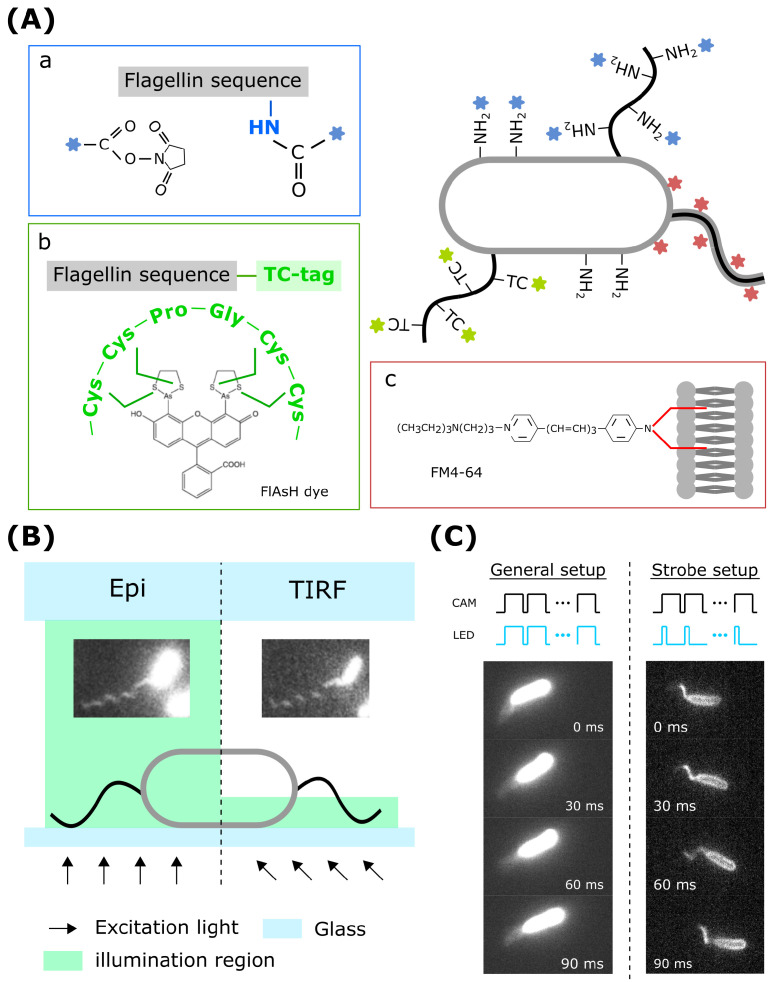
The observation of flagellar filaments. (**A**) Fluorescence labeling using fluorophore-linked succinimydyl (NHS) esters (box a), a TC-tag (box b), and a sheath (box c). (**B**) Schematics of the EPI and TIRF fluorescence imaging of a sheathed bacterium *V. alginolyticus* moving near the surface. The cell is labeled with FM 4-64. (**C**) For rapidly rotating flagellar filaments, long excitation times result in blurred images. Strobe illumination provides clear images of filament configurations. The cells are sheathed *A. fisheri* labeled with FM 4-64.

**Figure 3 biomolecules-15-00170-f003:**
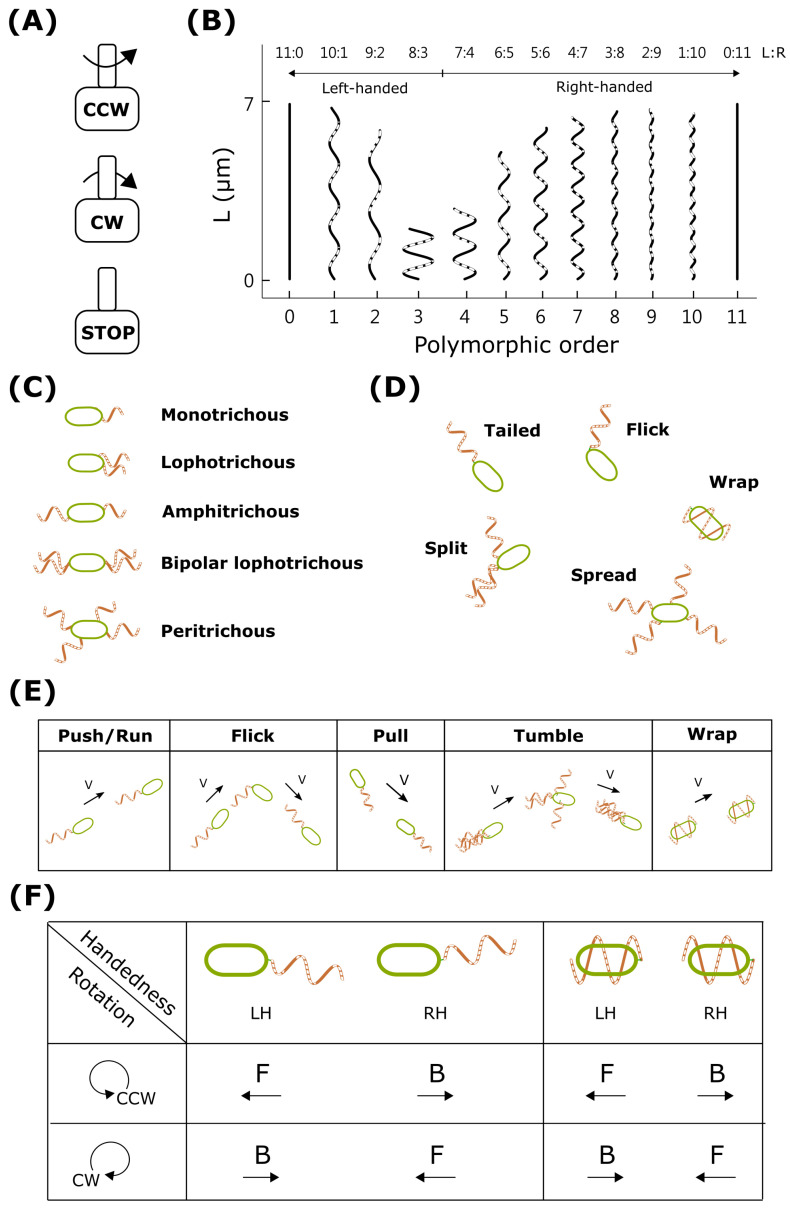
The decomposition of swimming states. (**A**) Flagellar motor rotation states: CCW, CW, and STOP. (**B**) Flagellar filament polymorphism. (**C**) Five major flagellar filament arrangements. (**D**) Flagellar configuration. (**E**) Swimming modes. (**F**) Run-mode swimming directions and swimming states.

**Figure 4 biomolecules-15-00170-f004:**
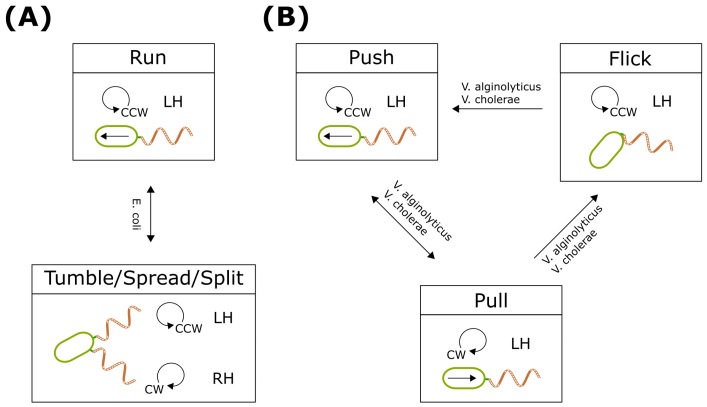
Kinetic model for swimming patterns. (**A**) Run-and-tumble. (**B**) Push–pull–flick.

**Figure 5 biomolecules-15-00170-f005:**
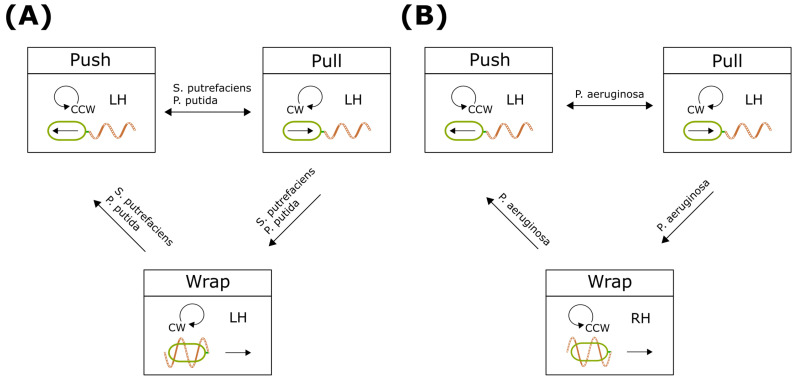
Kinetic model for swimming patterns. (**A**) Typical wrap-mode transition. The wrapping event occurs during the motor rotation transition from CCW to CW. (**B**) *P. aeruginosa* wrap-mode transition. The wrapping event occurs after the pull mode resumes CCW rotation. This is similar to the flick mode of *V. alginolyticus*.

**Figure 6 biomolecules-15-00170-f006:**
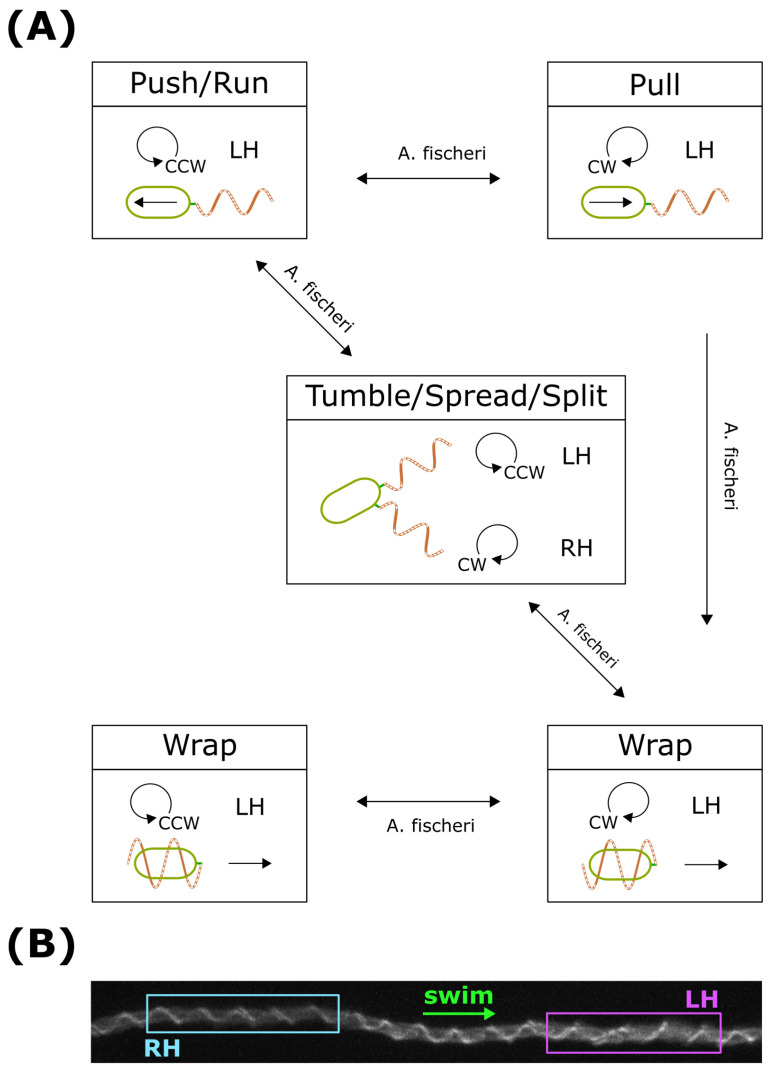
Kinetic model for swimming pattern in wrap mode. (**A**) *A. fischeri* swimming pattern with wrap mode. (**B**) The flagellar filament polymorphic transition during wrap mode is induced by flagellar motor switching. Reproduced with permission from CJLo [26]; published by Physical Review Research, 2024.

**Figure 7 biomolecules-15-00170-f007:**
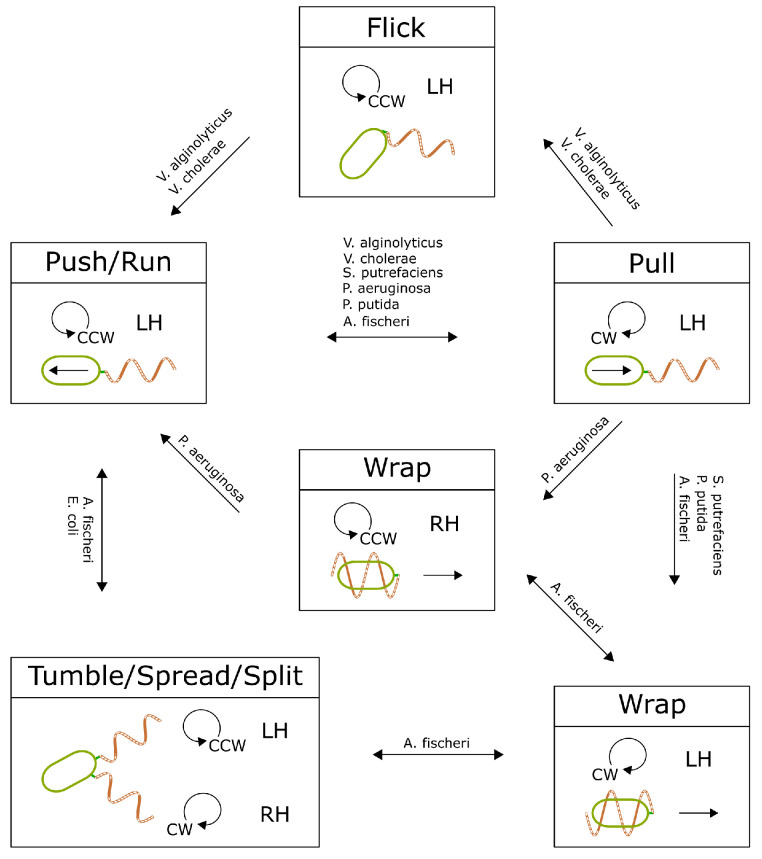
The collection of known bacterial swimming modes and possible transitions. It shows the limited routes of the transitions.

**Table 1 biomolecules-15-00170-t001:** The summary of the swimming mode, flagellar polymorphism, and motor rotation in different species.

Flagellar Arrangement	Specie	Swimming Mode	Flagellar Polymorphism	Motor Rotation	Environment	Ref.
Peritrichous	*E. coli*	Run	LH	CCW	MB	[14,78]
		Tumble	LH & RH	CCW & CW		
Monotrichous	*V. alginolyticus*	Push	LH	CCW	TMN	[6,8]
		Pull	LH	CW		
		Flick	LH	CCW		
Monotrichous		Push	LH	CCW		
	*V. cholerae*	Pull	LH	CW	M9MM	[80]
		Flick	LH	CCW		
Monotrichous		Push	LH	CCW		
	*P. aeruginosa*	Pull	LH	CW	MB (20% Ficoll400)	[12]
		Wrap	RH	CCW		
Monotrichous		Push	LH	CCW		
	*S. putrefaciens*	Pull	LH	CW	MB (10% Ficoll400)	[9]
		Wrap	LH	CW		
Lophotrichous		Push	LH	CCW		
	*P. putida*	Pull	LH	CW	TB:MB = 1:10	[16]
		Wrap	LH	CW		
Lophotrichous		Push	LH	CCW		
	*Burkholderia*	Pull	—	—	YG medium (surface)	[10]
		Wrap	LH	CW		
Lophotrichous		Push	LH	CCW		
	*A. fischeri*	Pull	LH	CW	TMN300 (0.89cP–1.94cP)	[26]
		Wrap	LH or RH	CW or CCW		
		Push	LH	CCW		
Amphitrichous	*C. jejuni*	Pull	—	—	MH medium (0.15–1% methylcellulose)	[81]
		Wrap	LH	CCW		

## Data Availability

No new data were created or analyzed in this study. Data sharing is not applicable to this article.

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
