# Peer review of "Decoding Bacterial Motility: From Swimming States to Patterns and Chemotactic Strategies"

_biomolecules, 2025, doi:10.3390/biom15020170_

Round 1

Reviewer 1 Report

Comments and Suggestions for Authors

The review by Zhuang and Lo describes the bacterial swimming patterns produced by bacterial flagellar rotation, filament polymorphism, and flagellar arrangement. The authors are frontrunners who have contributed to developing fluorescent staining of flagella. They have provided a clear summary of the recent methodology of fluorescent flagella staining and the new motility patterns revealed by these advancements. The text is well-written, and enough figures are given to explain the contents. Only some minor suggestions are listed below.

1. Fig.1: The two figures showing “bacterial random walk” enclosed by the broken line in the upper right corner appear small. I suggest that they be enlarged.

2. Fig. 2 (A) b: This should be FlAsH dye, but it was probably mislabeled as “FIAsH dye” in the cartoon.

3. Fig. 2(B): Change “Exitation light” to “Excitation light”.

4. L138-139: This sentence is the wrong description. CheA is auto-phosphorylated, not phosphorylated by CheW. CheA-P transfers P to CheY but does not lead to CheY being produced. Furthermore, the authors should describe that CheY-P is de-phosphorylated by CheZ to reduce the intracellular concentration of CheY-P, leading to the population of CW rotation. In relation to this, it is better to change the “W” to “Z” in Figure 1, upper-left (Signal transduction in E. coli).

5. L153: Probably better to rephrase, “the rotor from the proton-powered stator”.

6. L284: Remove “Edit here”.

7. Fig. 7: The title is the same as Fig.1 and is unsuitable for Fig. 7. Please reconsider it.

8. L381: Change “alpha epresents”to“alpha represents.

Author Response

The review by Zhuang and Lo describes the bacterial swimming patterns produced by bacterial flagellar rotation, filament polymorphism, and flagellar arrangement. The authors are frontrunners who have contributed to developing fluorescent staining of flagella. They have provided a clear summary of the recent methodology of fluorescent flagella staining and the new motility patterns revealed by these advancements. The text is well-written, and enough figures are given to explain the contents. Only some minor suggestions are listed below.

Response: We thank reviewer for the positive comments.

Comment 1: Fig.1: The two figures showing “bacterial random walk” enclosed by the broken line in the upper right corner appear small. I suggest that they be enlarged.

Response 1: The figures are enlarged. 

Comment 2: Fig. 2 (A) b: This should be FlAsH dye, but it was probably mislabeled as “FIAsH dye” in the cartoon.

Response 2: We are sorry about this typo. It has been corrected. 

Comment 3. Fig. 2(B): Change “Exitation light” to “Excitation light”.

Response 3: Corrected.

Comment 4. L138-139: This sentence is the wrong description. CheA is auto-phosphorylated, not phosphorylated by CheW. CheA-P transfers P to CheY but does not lead to CheY being produced. Furthermore, the authors should describe that CheY-P is de-phosphorylated by CheZ to reduce the intracellular concentration of CheY-P, leading to the population of CW rotation. In relation to this, it is better to change the “W” to “Z” in Figure 1, upper-left (Signal transduction in E. coli).

Response 4: We thank review’s critical comments. We are really sorry about this mistake and have corrected the paragraph and figures according to reviewer’s corrections.

Comment 5. L153: Probably better to rephrase, “the rotor from the proton-powered stator”.

Response 5: Rephrased.

Comment 6. L284: Remove “Edit here”.

Response 6: Removed.

Comment 7. Fig. 7: The title is the same as Fig.1 and is unsuitable for Fig. 7. Please reconsider it.

Response 7: We are really sorry about this mistake. It has been corrected. 

Comment 8. L381: Change “alpha epresents”to“alpha represents.

Response 8: Corrected. 

Reviewer 2 Report

Comments and Suggestions for Authors

In their paper, Zhuang and Lo give a comprehensive overview of the underlying mechanism of bacterial motility across a range of scales, from the flagellar motor to flagellar structure and arrangements, as well as how these structural features come together to manifest different motility strategies across species. It’s rare to have this range of information about bacterial locomotion gathered in one place, and as such, this review is useful to the community (in particular when the micromotility literature can be dense). I’m happy to recommend the publication of this paper in Biomolecules. I only have some smaller suggestions to clarify a few aspects of the presentation, which I list below.

Comments:

- Figure 1 is a bit confusing as constructed. There are two ‘modules’ noted within the bacterial motility system, but the signal transduction (chemotaxis) module does not seem to connect to anything. Surely it’s important for determining whether a random walk or a directed random walk is observed? Could this be made more explicit? If I’m mistaken and it indeed does not need to be connected to anything else in the Figure, perhaps it should be left out and only the locomotory apparatus should be focused on. 

- Line 52-53: Are these flagellar dimensions true across bacteria or for a particular species?

- In Figure 2B and C, by which method were the flagella labeled? This should be noted in the Figure caption.

- The states in Figure 3C are spelled incorrectly (same typo in all states)

Author Response

In their paper, Zhuang and Lo give a comprehensive overview of the underlying mechanism of bacterial motility across a range of scales, from the flagellar motor to flagellar structure and arrangements, as well as how these structural features come together to manifest different motility strategies across species. It’s rare to have this range of information about bacterial locomotion gathered in one place, and as such, this review is useful to the community (in particular when the micromotility literature can be dense). I’m happy to recommend the publication of this paper in Biomolecules. I only have some smaller suggestions to clarify a few aspects of the presentation, which I list below.

Response: We thank reviewer's positive comments.

Comments:

Comment 1: Figure 1 is a bit confusing as constructed. There are two ‘modules’ noted within the bacterial motility system, but the signal transduction (chemotaxis) module does not seem to connect to anything. Surely it’s important for determining whether a random walk or a directed random walk is observed? Could this be made more explicit? If I’m mistaken and it indeed does not need to be connected to anything else in the Figure, perhaps it should be left out and only the locomotory apparatus should be focused on. 

Response 1: We thank reviewer for this important comment. As reviewer point out, bacterial chemotaxis integrates signal transduction with locomotion. In this review, we focus on the locomotion apparatus but it cannot be mention chemotaxis without the signal transduction. We made Figure 1 more clear for this review by changing the focused locomotion part in solid-lines and the signal transduction parts in gray-dash lines.

Comment 2: Line 52-53: Are these flagellar dimensions true across bacteria or for a particular species?

Response 2: We thank reviewer for this question. We are sorry about the typo of the flagellar outer diameter 24 nm quoted from ref [13]. As reviewer point out, the flagellar dimensions are varied from species. Here we would like to give the readers the impression of the typical dimension of the flagellar filament. We have rephrased the sentence for clarity. 

Comment 3: In Figure 2B and C, by which method were the flagella labeled? This should be noted in the Figure caption.

Response 3: Information added.

Comment 4: The states in Figure 3C are spelled incorrectly (same typo in all states)

Response 4: Corrected

Reviewer 3 Report

Comments and Suggestions for Authors

This is a comprehensive and timely review of an interesting subject. It is very thorough and exteremely well-written, especially considering that I presume the authors are not native English speakers. My only substantive criticism if that the last sentence of the Conclusion is rather vague, as I noted in my detailed comments. 

Here are my editorial suggestions. They are mostly minor, and all are designed to make the text and figures a bit more accurate, streamlined, and in smooth-flowing style.

     1)      Line 53. Delete “approximately.”

2)      Line 113. Replace “up to kilohertz” with “in the kilohertz range.”

3)      Line  120. Delete “duration.”

4)      Figure 2 legend. Rewrite as “The observation of flagellar filaments. (A) Fluorescence… imaging of bacteria moving on a surface. (C) For rapidly rotating flagellar filaments, long excitation times result in blurred images. Strobe illumination provides clear images of filament configurations.”

5)      Line 134. Insert comma. “…(Figure 3A), and the….”

6)      Line 150. Insert the “…in the Bacillus subtilis BFM….”

7)      Line 159.  Write as “…of the BFM rotation: CCW/CW/STOP.”

8)    Line 161. Write as “…is a helical propellor….”

9)    Line 170. Delete “…through molecular and theoretical analyses….”

10) Line 172. Rewrite as “…switching and produces many swimming patterns….”

11)  Line 173. Rewrite as “…such as the 3D structure of flagellin, ionic strength….”

12)  Lines 176-177. The arrangement of flagella on the cell surface plays a crucial role in motility and chemotaxis and can be classified into five major groups (Figure 3C)….”

13)  Line 198. Write as “…between the RH and LH states.”

14)  Line 199. Replace “intricate” with “complex.”

15)  Line 202. Rewrite as “The primary swimming modes are push, pull, tumble, flick, and wrap. These actions are the combined result of the direction of flagellar rotation, the filament configuration, filament polymorphism, and the arrangement of flagella on the cell surface.”

16)  Lines 205-207. Rewrite as “The cell enters the ‘tumble’ mode by switching the rotation of one or more motors to CW, changing some filaments to RH and producing a split configuration.”

17)  In Figure 3c, all five “...trichours” should be “...trichous.”

18)  In Figure 3 legend. Should be “(A) Flagellar…” and “(C) Five….”

19)  Lines 244-245. Rewrite as “…that ensures brief and efficient tumbling that enables bacteria to explore their environment optimally to perform chemotaxis….”

20)  Lines 266-267. Rewrite as “…and prolongs CCW rotation when swimming in a favorable direction [6]. V. alginolyticus can thus bias its swimming direction very quickly [59,60].”

21)  Line 287. Should the first ‘pull’ be a ‘push’?

22)  Line 298. Delete “of bacteria.”

23)  Line 302. Replace “initiate due to” with “be caused by.”

24)  Line 307. The meaning of “over-whirling” is unclear. Choose a different description.

25)  Lines 310-311. Rewrite as “The degree of reorientation of the swimming direction depended on the length of motor stoppage between periods of motor rotation.”

26)  Line 314. Rewrite as “…depending on the direction of motor rotation.”

27)  Lines 318-319. Rewrite as “…reveal that the filament of P. aeruginosa assumes a right-handed helix in the wrap mode, unlike its typical….”

28)  Figure 6 legend. Rewrite as “…pattern in the wrap mode.”

29)  Line 329. Rewrite as “…dynamic polymorphic transition of the filament during….”

30)  Line 331. The “diminishes” should be “diminish.”

31)  Lines 338-340. Rewrite as “During the wrap mode, filament transformations induced by motor switching extend backward swimming to generate long runs (Figure 6B). If the synchronization of motor rotation is disrupted, the….”

32)  Line 345. The “high viscous” should be either “highly viscous” or “high viscosity.”

33)  Line 47-348. Rewrite as “produces lateral flagella to…bacteria that lack a secondary….”

34)  Line 364. Replace “integrating” with “which integrates.”

35)  Lines 371-372. “…inhibit the production of CheY-P…to continue running toward….”

36)  Line 381. Should be “and a represents the….”

37)  Line395. Should be “…to the scale of bacterial cells or flagella.”

38)  Lines 399-400. Rewrite as “Considering the complexity and adaptive value of bacterial motility, it is likely that additional swimming states await discovery.”

39)  Line 402. Rewrite “As more patterns of bacterial motility are….”

40)  Line 409. The Conclusion ends with “…could inform the design of novel microscopic active agents.” What is meant by this statement? Genetically engineered bacteria? Miniature robotic devices that function like bacteria? Be more specific and concrete in the final sentence.

Author Response

This is a comprehensive and timely review of an interesting subject. It is very thorough and exteremely well-written, especially considering that I presume the authors are not native English speakers. My only substantive criticism if that the last sentence of the Conclusion is rather vague, as I noted in my detailed comments. 

Here are my editorial suggestions. They are mostly minor, and all are designed to make the text and figures a bit more accurate, streamlined, and in smooth-flowing style.

Response: We thank reviewer's positive comments and careful editorial suggestions. We really appreciate these assistance to make this reivew better.

Comment 1) Line 53. Delete “approximately.”

Response 1) Deleted.

Comment 2) Line 113. Replace “up to kilohertz” with “in the kilohertz range.”

Response 2) Replaced.

Comment 3) Line  120. Delete “duration.”

Response 3) Deleted.

Comment 4) Figure 2 legend. Rewrite as “The observation of flagellar filaments. (A) Fluorescence… imaging of bacteria moving on a surface. (C) For rapidly rotating flagellar filaments, long excitation times result in blurred images. Strobe illumination provides clear images of filament configurations.”

Response 4) Corrected.

Comment 5) Line 134. Insert comma. “…(Figure 3A), and the….”

Response 5) Corrected.

Comment 6) Line 150. Insert the “…in the Bacillus subtilis BFM….”

Response 6) Inserted.

Comment 7) Line 159.  Write as “…of the BFM rotation: CCW/CW/STOP.”

Response 7) Corrected.

Comment 8) Line 161. Write as “…is a helical propellor….”

Response 8) Corrected.

Comment 9) Line 170. Delete “…through molecular and theoretical analyses….”

Response 9) Deleted.

Comment 10) Line 172. Rewrite as “…switching and produces many swimming patterns….”

Response 10) Corrected.

Comment 11) Line 173. Rewrite as “…such as the 3D structure of flagellin, ionic strength….”

Response 11) Corrected.

Comment 12)  Lines 176-177. The arrangement of flagella on the cell surface plays a crucial role in motility and chemotaxis and can be classified into five major groups (Figure 3C)….”

Response 12) Corrected.

Comment 13)  Line 198. Write as “…between the RH and LH states.”

Response 13) Corrected.

Comment 14)  Line 199. Replace “intricate” with “complex.”

Response 14) Corrected.

Comment 15)  Line 202. Rewrite as “The primary swimming modes are push, pull, tumble, flick, and wrap. These actions are the combined result of the direction of flagellar rotation, the filament configuration, filament polymorphism, and the arrangement of flagella on the cell surface.”

Response 15) Corrected.

Comment 16)  Lines 205-207. Rewrite as “The cell enters the ‘tumble’ mode by switching the rotation of one or more motors to CW, changing some filaments to RH and producing a split configuration.”

Response 16) Corrected.

Comment 17)  In Figure 3c, all five “...trichours” should be “...trichous.”

Response 17) Corrected.

Comment 18)  In Figure 3 legend. Should be “(A) Flagellar…” and “(C) Five….”

Response 18) Corrected.

Comment 19)  Lines 244-245. Rewrite as “…that ensures brief and efficient tumbling that enables bacteria to explore their environment optimally to perform chemotaxis….”

Response 19) Corrected.

Comment 20)  Lines 266-267. Rewrite as “…and prolongs CCW rotation when swimming in a favorable direction [6]. V. alginolyticus can thus bias its swimming direction very quickly [59,60].”

Response 20) Corrected.

Comment 21)  Line 287. Should the first ‘pull’ be a ‘push’?

Response 21) Corrected.

Comment 22) Line 298. Delete “of bacteria.”

Response 22) Deleted

Comment 23)  Line 302. Replace “initiate due to” with “be caused by.”

Response 23) Replaced.

Comment 24)  Line 307. The meaning of “over-whirling” is unclear. Choose a different description.

Response 24) We thank reviewer for this comment. “Over-whirling” is the mode observed by simulation in the ref [67]. This mode is similar to the flick mode with higher hook-bending angle. We re-rephrase it into “flick-like mode for clarity.

Comment 25)  Lines 310-311. Rewrite as “The degree of reorientation of the swimming direction depended on the length of motor stoppage between periods of motor rotation.”

Response 25) Rewrited.

Comment 26)  Line 314. Rewrite as “…depending on the direction of motor rotation.”

Response 26) Rewrited.

Comment 27)  Lines 318-319. Rewrite as “…reveal that the filament of P. aeruginosa assumes a right-handed helix in the wrap mode, unlike its typical….”

Response 27) Rewrited.

Comment 28)  Figure 6 legend. Rewrite as “…pattern in the wrap mode.”

Response 28) Rewrited.

Comment 29)  Line 329. Rewrite as “…dynamic polymorphic transition of the filament during….”

Response 29) Rewrited.

Comment 30)  Line 331. The “diminishes” should be “diminish.”

Response 30) Corrected.

Comment 31)  Lines 338-340. Rewrite as “During the wrap mode, filament transformations induced by motor switching extend backward swimming to generate long runs (Figure 6B). If the synchronization of motor rotation is disrupted, the….”

Response 31) Corrected.

Comment 32)  Line 345. The “high viscous” should be either “highly viscous” or “high viscosity.”

Response 32) Corrected.

Comment 33)  Line 47-348. Rewrite as “produces lateral flagella to…bacteria that lack a secondary….”

Response 33) Rewrited.

Comment 34)  Line 364. Replace “integrating” with “which integrates.”

Response 34) Replaced.

Comment 35)  Lines 371-372. “…inhibit the production of CheY-P…to continue running toward….”

Response 35) Corrected.

Comment 36)  Line 381. Should be “and a represents the….”

Response 36) Corrected.

Comment 37)  Line395. Should be “…to the scale of bacterial cells or flagella.”

Response 37) Corrected.

Comment 38)  Lines 399-400. Rewrite as “Considering the complexity and adaptive value of bacterial motility, it is likely that additional swimming states await discovery.”

Response 38) Corrected.

Comment 39)  Line 402. Rewrite “As more patterns of bacterial motility are….”

Response 39) Corrected.

Comment 40)  Line 409. The Conclusion ends with “…could inform the design of novel microscopic active agents.” What is meant by this statement? Genetically engineered bacteria? Miniature robotic devices that function like bacteria? Be more specific and concrete in the final sentence.

Response 40) We thank reviewer for this important suggestions. The attempt of the final sentence is to extend our fundamental science understanding toward potential applications. We have rephrased it according to reviewer’s suggestions.